# Area efficient camouflaging technique for securing IC reverse engineering

**Md. Liakot Ali**[1]☯*, **Md. Ismail Hossain**[1‡], **Fakir Sharif Hossain**[2‡]

**1** Institute of Information and Communication Technology, Bangladesh University of Engineering and Technology, Dhaka, Bangladesh, **2** Department of Electrical and Electronic Engineering, Ahsanullah University of Science and Technology, Dhaka, Bangladesh

☯ These authors contributed equally to this work.
‡ MIH and FSH are authors also contributed equally to this work.
* liakot@iict.buet.ac.bd

**Data Availability Statement:** All relevant data are within the manuscript.

**Funding:** The authors received no specific funding for this work.

## Abstract

Reverse engineering is a burning issue in Integrated Circuit (IC) design and manufacturing. In the semiconductor industry, it results in a revenue loss of billions of dollars every year. In this work, an area efficient, high-performance IC camouflaging technique is proposed at the physical design level to combat the integrated circuit's reverse engineering. An attacker may not identify various logic gates in the layout due to similar image output. In addition, a dummy or true contact-based technique is implemented for optimum outcomes. A library of gates is proposed that contains the various camouflaged primitive gates developed by a combination of using the metal routing technique along with the dummy contact technique. This work shows the superiority of the proposed technique's performance matrix with those of existing works regarding resource burden, area, and delay. The proposed library is expected to make open source to help ASIC designers secure IC design and save colossal revenue loss.

## Introduction

Reverse engineering (RE) is one of the primary concerns and long-standing problems to governments, militaries, and industries all over the world [1]. It covers objects from as large as aircraft down to the smallest Integrated Circuit (IC), through which an object is examined to gather a complete understanding of its construction and/or functionality. It is recognized worldwide and has many noble purposes: failure analysis and defect identification, detection of counterfeit products, recovery of manufacturing defects, Confirmation of Intellectual property, Hardware Trojan detection, education, and research, etc. [2–4]. However, now globally, it has become a significant concern and anxiety for different stakeholders since it can be used for different malicious purposes such as cloning, pirating, or counterfeiting a design and developing an attack or inserting a hardware Trojan. The devastating effect of RE is already proven in the history of world war II, and the Vietnam war [5, 6]. Currently, we live in the era of industrial revolution 4 (IR4), where automation and artificial intelligence are everywhere. However, to implement automation, the most fundamental component of any electronic system is IC.

**Competing interests:** The authors have declared that no competing interests exist.

Due to the blessings of continuous improvement of IC technology, millions of transistors are being integrated into a single chip and are facilitating to implementation of the full functionalities of a system which is known as a system on a chip (SoC) [7].

On the other hand, continuous increasing design complexity and high design cost have led to the globalization of IC design and fabrication. All phases of supply chain rouge can perform different malicious activities such as IP infringement, IC counterfeiting, overbuilding, hardware Trojans, side-channel attacks. It can cause serious security and economic concerns in the semiconductor industry, resulting in billions of dollars lost each year [8–10]. A major enabler of these malicious and unlawful activities is reverse engineering (RE), where the rogues usually exploit the advantages of commercially available RE tools that were developed for honest purposes as fault analysis, chip testing and verification. Circuit camouflaging techniques have been proven effective to combat the reverse engineering attacks [11–13].

Reverse engineering attacks recover the original netlist through scanning electron microscopy (SEM) images [14, 15]. It is usually applied in combinational logic of an application-specific integrated circuit (ASIC) and proactively hides the layout information of intellectual properties (IPs), intending to make RE exponentially more difficult [16]. Specifically, it hides IC's design information by replacing some conventional logic gates with specially designed camouflaged cells, in which different types of camouflaged gates have been configured to perform one of the multiple functionalities while maintaining an identical look to RE attackers [17]. Therefore, while the attacker performs top-down reverse engineering, it is difficult for them to identify the camouflaged gates' actual functionalities or challenging to identify the actual netlist resulting in failing of RE [18].

This paper focuses on research based on area efficient camouflaging techniques and the development of a library of camouflaged primitive logic gates using the said technique for application specific IC (ASIC) designers.

## Related works and motivation

To secure the ICs and IPs from RE is now a hardcore problem in the semiconductor industry that must be prevented otherwise, it turns towards many devastating consequences in different sectors of a country [1]. To combat the RE in the semiconductor industry, circuit camouflaging has become one of the hottest research topics in hardware security since 2012. Numerous techniques are in the literature to secure a design through camouflaging. In [14], the authors presented two camouflaging techniques to develop a secure cell library. The cell library is secured through an AND-tree structure, and a combined strategy is adopted to reduce the cost. They show the resistance of the SAT-based attack. In the literature, IC camouflaging techniques are mainly three types: post-silicon, cell level, and gate-level netlist [19–21]. Gate level netlist-based methods use an algorithm to resilience circuit elements for RE attacks with sufficient hardware overhead. By changing the post-silicon state's dopant polarity, transistors are always kept ON or OFF state to protect from the RE. A dummy contact-based method is proposed in [22] to control two adjacent layers.

In a traditional way, camouflaged cells' implementation is placed in device level with ambiguous gates [16, 23]. This way requires a fully trustworthy fab process since the foundry might perform the obfuscation. A group of researchers present multiplexer based obfuscation [24, 25] where others rely on threshold dependent implementation [20, 26, 27]. In [23], authors show the extra power, area, and performance parameters significantly due to the process design. As technology scales down to 65nm, the parameters cost higher [28]. A low-cost camouflaging technique is proposed in [29]. Authors present a technique even if an untrusted fab prevails through back-end-of-line security with optimization of cost. Using a pull-down

resister, the technique can control the current flow which results a reduction of average power and delay. The resulting delay is a little bit less than that of conventional gate design as the gate has a smaller voltage swing from logic-1 to logic-0. A popular technique named printed electronics is presented in [30]. It delivers mechanical flexibility with low cost and on-demand fabrication. However, the method is vulnerable in RE attacks as the recent demonstration is performed. To overcome this, the authors propose printed camouflaged logic cells with negligible overhead. In [31], a cyclic obfuscation under different cycle conditions is proposed to show the complexity level of RE with an exponentially increasing time.

A different functionality of camouflaged cells can have obtained by changing vias or doping level of the semiconductor [14, 32–36]. Only an optimized percentage of camouflaged cells are inserted in the design to deliver maximum efforts of decamouflaging with a negligible overhead [16, 37]. Therefore a significant number of research works is in the literature trying to reduce the overhead by increasing the security level.

True or dummy contact-based approach is the most popular method to make camouflaged cells as this technique works in alternate ways between two contacts based on the requirements [38]. In this process, two contacts alternatively act like active or inactive. Instruction needs to provide the fabrication lab to apply a special layer so that contacts can decide automatically to be activated or inactivated. Another way of making camouflaged cells is termed as 'covert gate' that leverages doping and dummy contacts to create camouflaged cells [39]. Few other methods are also popular, like SRAM based camouflaging and filler cell based camouflaging [40]. These methods make the overall performance slower than common methods as they need extra layers or extra cells along with the logic gates. The layout design technique is the most effective method and better in case of performance overhead.

An open source library of various camouflaged gates of different combinational logic gates using true or dummy contact based technique may help the ASIC designer to secure their design from RE and may save huge revenue loss of manufactures [41]. However, the ASIC designer needs to know each logic gate's performance matrix in terms of resource burden, power, and delay so that they can choose the optimum one. Therefore, a considerable scope of research in the said direction is required [42].

In the conventional NAND and NOR gates, a similarity in design to make confusion from the top view can be implemented to make output connection in the bottom line. It may create a short contact with other designs while merging them side by side vertically. However, it is standard practice to design Ground in the bottom and $V_{DD}$ in the upper line so that the design can be merged with the other design side by side to reduce area and increase performance [43]. Similarly, the dummy contact-based implementation increases area as several connections are done by metal routing, resulting in misusing the poly's metal.

We apply layout optimization with a dummy technique to avoid area misuse and maintain industry standards in this work. We implement design tricks while routing with metal 1 and metal 2 so that area remains almost the same as the conventional design and falls secure with dummy based technique. The proposed approach is to design camouflaged cells at the layout level while minimizing the power and area overhead. The proposed method rests on designing logic gates by implementing layout techniques and developing an open-source standard cell library so that ASIC designers can use this library for designing chips. The main contributions of this work are summarized below.

- To design universal logic gates with modified layout based technique with the dummy contact methodology.

- To deliver secure chip with small area overhead, better performance with less power consumption.

- To develop a full library that would be used to chip design and development.

The remaining Sections of this paper are organized as follows. A discussion on IC camouflaging, their effects, benefits, and applications in the IC design industry are presented in the following Section. In the third Section, a threat model of the proposed approach is discussed. The fourth Section briefly discusses how the approach is made towards the objective and how the implementation is with the proposed design. The proposed method is presented in the fifth Section, and experimental results with comparisons are shown in the sixth Section. Finally, we summarize the work in the conclusion Section.

## IC camouflaging

Different types of IC camouflaging are reported in the literature. This process is divided into two main sub-processes: Camouflaged IC designing and Reverse Engineering. The methodology and fundamental process of using IC camouflaging are presented in detail in [16].

### Camouflaged IC designing

The first step of the entire process starts with basic IC designing. The method of making an IC can be classified into two more sub-categories. First, it needs to be designed. Further, in the second stage, it needs to be manufactured. The design process is again subdivided into front-end and back-end processes. They are again classified in different steps: specification, register transfer logic (RTL) design, functional verification, synthesis, and physical design. Specification means writing the functionality of the IC and defining its microarchitecture. These specifications are further programmed in a hardware description language (HDL) like Verilog in the RTL design.

In the functional verification phase, the RTL design's correctness and, the quality and completeness of the testing process are ensured. Back-end processes include synthesis and physical design. After being verified, the HDL code is converted into a digital circuit using logic gates and flip-flops only. The digital circuit consists of a list of logic gates that are connected appropriately to perform the IC's designed functionality. This circuit diagram of the final IC comprising of logic gates is known as netlist.

Further, when the design goes into synthesis, it uses the standard cells and uses the camouflaged cells. The design creates a layout in the physical design part, and the layout consists of both standard and camouflaged cells. This layout is sent to fabrication in the foundry. From the defenders' point of view, some decisions on camouflaged cells are made. The number of camouflaged cells the design affords with the structural usage and in what part of the design it may use these camouflaged cells.

These decisions ensure the security that the design attains using camouflaged cells and have implications for the area, power, and delay overhead that the camouflaged cells incur. Usually, a designer cannot camouflage all the existing gates being present while designing due to considering the factors mentioned above- area, power, and delay overhead. After all the procedures of designing are completed, the manufactured IC is ready for users.

### Reverse engineering of camouflaged process

From the attacker's point of view, the attacker takes the camouflaged chip and depackages it by using some corrosive chemicals. The epoxy packaging of the chips is removed and thus, exposing the dies. Further, the delayering of each layer, including diffusion, poly, and metal is done. The lower metal layers' delivery is more challenging than delaying the higher metal layers because of their relatively higher thickness.

Using an optical microscope, the imaging of the top view of the chip layers is conducted. This image displays all the metal routing, pins, and conducts presented in each layer. This imaging process can also be carried by using a single electron microscope as well.

The last stage of a reverse engineering technique is the extraction of netlist. For completing this stage successfully, the attacker may use the tools Degate or Chipworks. However, after extracting the netlist, the attacker may find some ambiguity due to the camouflaged cells' presence.

The attacker may not distinguish whether the used gate is NAND or NOR under the microscope because the contacts might be the true or dummy contact. Therefore, the attackers may not be able to know all the functionalities of the camouflaged cells. Therefore, an attacker may not reverse engineering on camouflaged cells, but they can do so to standard cells.

## Threat model

The design and layout phases are assumed to be trusted. The fab and end-users are considered untrusted. The camouflaged cells are designed in the design house in the layout phase. The fab has full access to the design layout; however, it may not access the back-end of the line. As the fab line's front-end has a direct contrast on camouflaged cells due to the front-end of line centric technique, the fab has to be trusted in the front-end of the line. To secure the back-end of line layers consisting of dummy vias and wires from fab's adversaries, we required a trusted back-end of the line facility.

The intruders intend to perform reverse engineering on the physical chip to identify the back-end line layers to reconstruct the original netlist and its IPs. We assume that the intruder cannot directly infer the secret back-end of line mappings but to the front-end of line layers. In addition, we consider the intruder has complete knowledge of the proposed approach, designed tools, and IPs. However, they cannot reconstruct due to randomized selection steps taken throughout our method.

## Basic of the proposed camouflaged IC design

In this work, a technique is proposed at the physical design level to reduce circuit functionality recognition through image processing methodology. An attacker may not identify various logic gates in the layout due to similar images yielding. Besides, a dummy or true contact-based technique is implemented for the optimum output. A library of gates is proposed containing different camouflaged primitive gates developed by combining the metal routing technique and the dummy contact technique. This work shows the performance matrix of implementation of the library of gates in XOR logic in terms of resource burden, area, and delay compared to conventional primitive gates. The proposed library is expected to make open source to help ASIC designers secure IC designing and save colossal revenue loss.

A high-level illustration is shown in Fig 1 displaying the idea of the implementation. Initially, universal gates are chosen to implement the camouflaging technique that would be feasible also for industrial design. Therefore, NAND and NOR gates are designed and simulated conventionally. After the simulation layout of these gates is done conventionally, modifications are done to introduce camouflaging technique. After the modifications, both gates have become almost similar perspectives of different layers and routing patterns. Incorporating with the true or dummy contact-method, both gates are 100% identical from the top view. For both gates, SEM image processing method delivers the same result.

For the initial implementation of camouflaging concept, two universal logic gates NAND and NOR are selected. The schematic and layout of conventional logic gates are designed and verified. The camouflaged logic gates of these two gates are drawn in a camouflaged way,

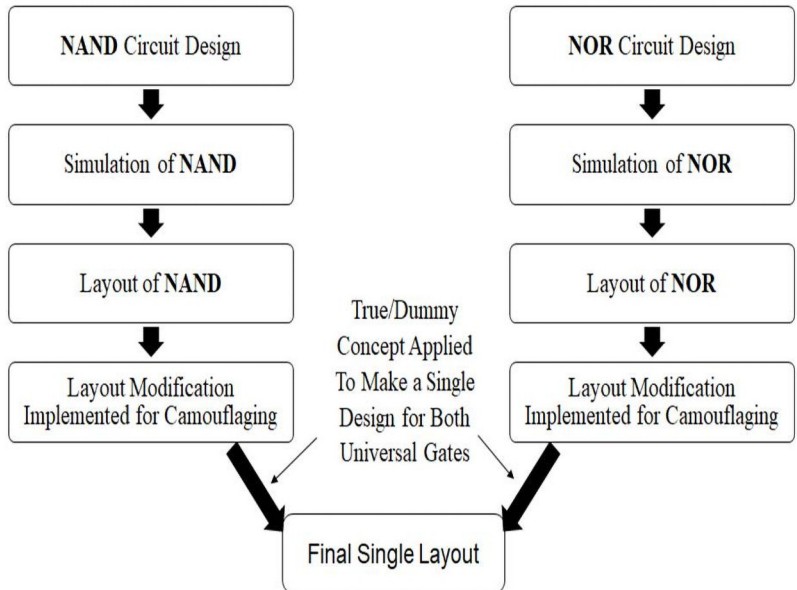

**Fig 1. Proposed IC camouflaging technique utilizing true or dummy contact method.**

applying modified routing. The routing modification takes for both metal 1 and metal 2. The XOR gate is designed using the camouflaged gates to measure the performance differences. As it may difficult to identify the optimization in NAND and NOR gates due to the small layout, an enormous XOR gate is selected to make the differences visible.

## Proposed universal gate design

The proposed method is to develop a library of universal gates. The obfuscation is done with designing the metal routing so that only via contacts position remains different. Except for vias, all the other layer positions and designs are the same in both NAND and NOR gates. Therefore, if the dummy contact technique is used while fabricating the design it shows almost similar configuration for both NAND and NOR gates which may sufficiently difficult to identify the difference in appearance of two gates. A detail procedure of designing universal gates is presented in Fig 2. The execution procedure delivers the camouflaged cells applying the proposed method.

The proposed logic gates layout is drawn in a camouflaged way by applying different metal routing techniques. The semiconductor design industry is suggested to use the lowest allowable metal layers while designing standard cells to overcome the extra delay issue and slow performance. In the proposed design, metal 1 and metal 2 are used for routing.

Considering the dummy contact technique, both NAND gate and NOR gate can be designed with a single gate. With the camouflaging technique modifications are done in layout. The application of dummy contact concept creates the whole design in a unified shape.

## Description of design steps

Algorithm-1 shows NAND and NOR gates designing with the proposed flow under different conditions. It includes transistors of PMOS and NMOS and all gates of NAND, NOR and XOR. A schematic is designed to check the functionality in the RTL by running a simulation. Based on the simulation results, a physical layout is designed caring physical verification and

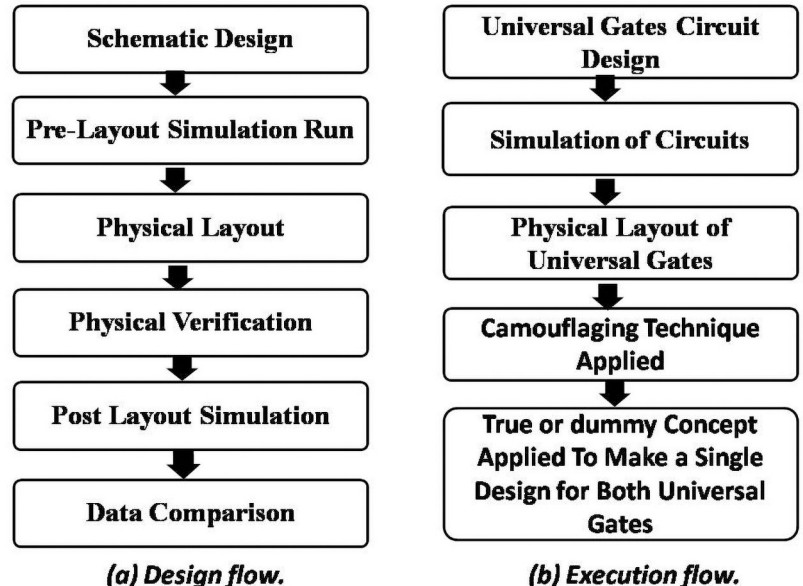

**Fig 2. Flowchart of the design and execution steps.**

RC extraction. Each input of the design schematic is simulated to extract information. Based on the extracted data physical layout is performed and verified. Meeting upon the condition of optimum layout, parasitic camouflaged cells are placed in the layout. This process repeats until the verification meets the proper functionality.

**Algorithm 1**: Camouflaging cells in layout modification

```
Input: Design schematic of gates;
Output: Camouflaged single layout;
while Simulation = True do
   Run simulation to extract information with layout;
   if Physical verification meets then
     Physical layout design;
     Physical verification;
     RC extraction;
     Camouflaged cells;
   else
     Simulation = False;
   end
end
Camouflaged cells making a single gate;
```

## Layout optimization

In the conventional design, the similarity between NAND and NOR gates in all layers exists. However, only the via connection positions are different in the design. In the conventional design, the metal 1 routing is used only. As there is no metal 2 routing exists it is possible to identify them separately. Due to metal 1 bending, it is recognizable of the series connection and parallel connections of the MOS. In the proposed design, using both metal 1 and metal 2 in different bendings, it is difficult to identify gates as the routings are almost similar. If the intruder tries to detect vias with precision, only in that case it would be possible to distinguish them. This identification can be void if we incorporate the dummy contact approach with the

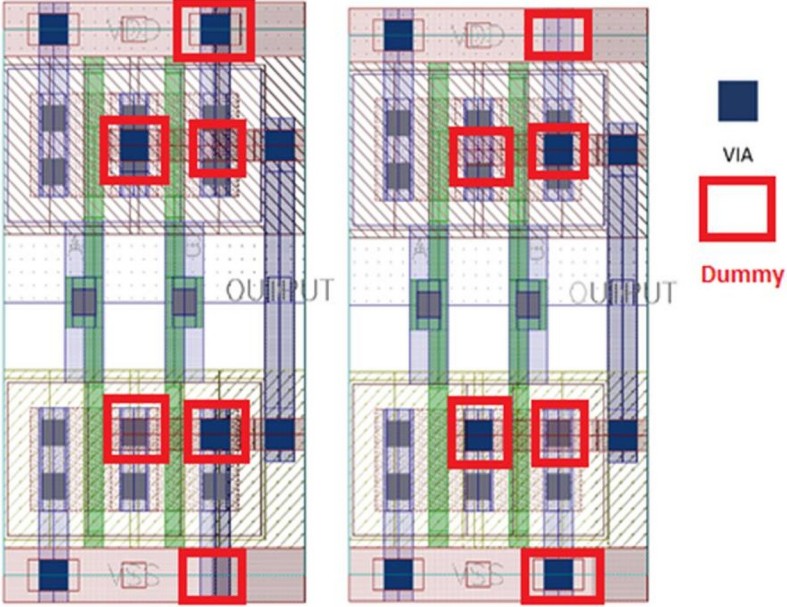

**Fig 3. Proposed method incorporating with dummy concept.**

proposed metal routing techniques. The dummy contact based camouflaging technique is applied with a modification in the layout of the design. Fig 3 shows the proposed technique.

The proposed method delivers more compact design considering 1) modifications in the layout design and 2) incorporating with the dummy contact technique. The proposed approach is better in area overhead than applying only the dummy contact method as the dummy contact-based approach utilizes a wider area to implement. This is because each input line is connected to drain voltage, $V_{DD}$ and Ground. In between the connections the dummy contacts are placed. If one contact is true then other one is disconnected or dummy and vice versa. Therefore, in every case, multiple contacts are required resulting in area overhead.

According to the dummy technique, if the selection lines $I_1$, $I_2$, $I_3$, and $I_4$ are the inputs noted by A and B and the output line is indicated by Y, the Boolean function of a camouflaged cell can be written as Eq (1).

$$Y = (A.B)\mathbf{I_1} + (A.B)\mathbf{I_2} + (A.B)\mathbf{I_3} + (A.B)\mathbf{I_4} \tag{1}$$

There can be 16 possible combinations of two inputs and one Boolean output function. If the selection values of $I_1$, $I_2$, $I_3$ and $I_4$ be 1110, the function of camouflaged cell becomes as Eq (2).

$$Y = (A.B) + (A.B) + (A.B) = (A.B) \tag{2}$$

Therefore, the camouflaged cell performs as a NAND gate. The dummy contacts are placed at the fabrication time as a skinny layer and disconnected after implementation. The top view displays the connections that exist but, in reality, are not connected. Therefore, it is not easy to figure out the exact design through SEM Images.

Both NAND and NOR gates can be designed with a single gate based on the proposed method. Getting NAND or NOR gate, the thin layer can be disconnected physical connections via where necessaries.

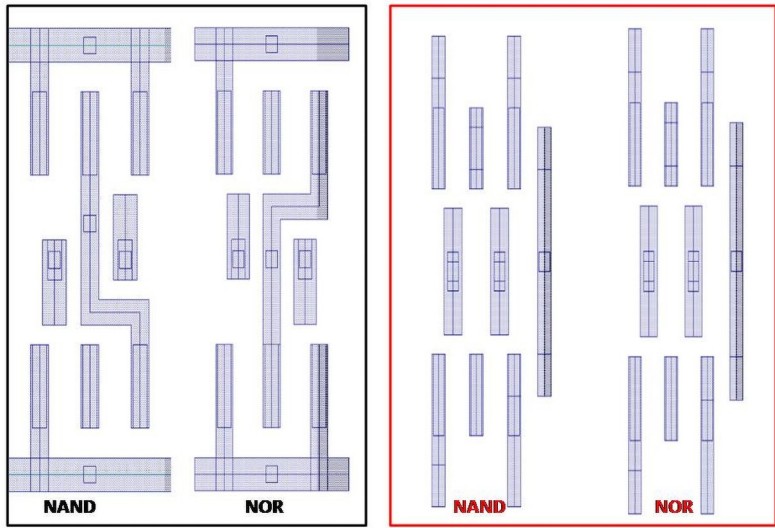

**(a) Conventional gates metal-1 routing**     **(b) Proposed gates metal-1 routing**

**Fig 4.** (a) Conventional gates metal routing and (b) proposed camouflaged gates metal 1 routing.

Fig 4, it is clear as we can see only the metal that connects all nodes. For the PMOS, it is visible that PMOS are in parallel and NMOS are in series connections. Then, it is possible to identify which is NAND or NOR. Metal 1 bending connection depicts the link. That bending is removed in the proposed design.

Fig 5 shows the straight connection of metal 2. Internal connection is done with metal 2 resulting in no bending of metal. It delivers a secure design from intruders.

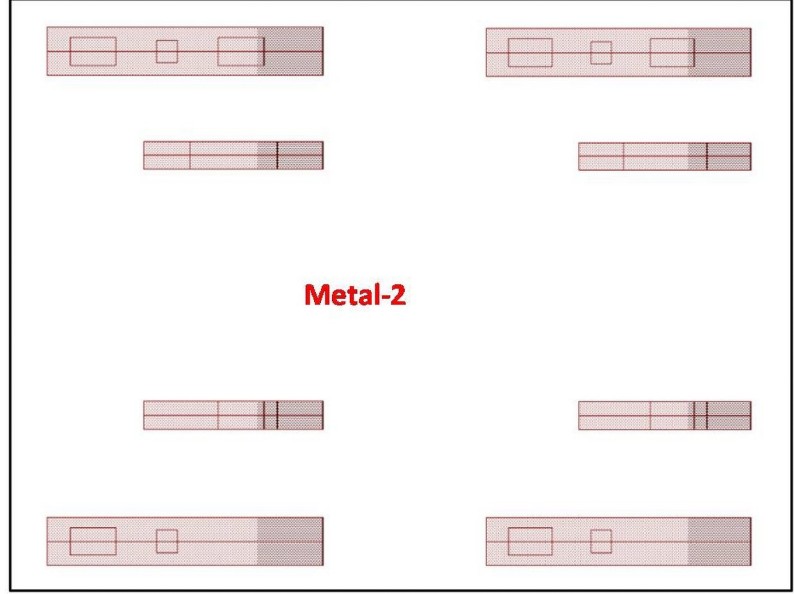

**Fig 5. Proposed camouflaged gates metal 2 routing.**

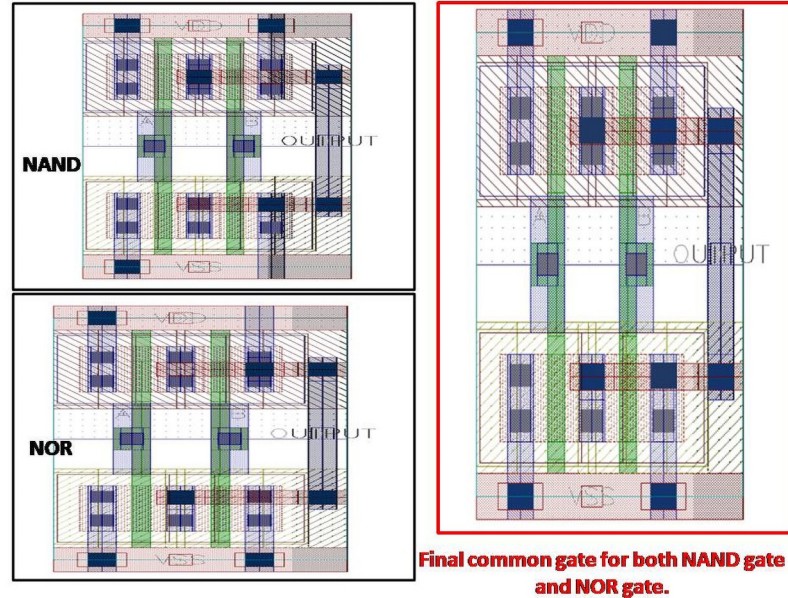

**Fig 6. Proposed camouflaged NAND and NOR gates.**

## Camouflaged NAND and NOR gates

Fig 6 shows the proposed camouflaged NAND and NOR gates and their single representation. These designs are done with modifications in the metal level routing. The blue color layers are metal 1 routing, and the red color layers are metal 2. In the conventional design, it is possible to connect these nodes with only metal 1. There are two metal 2 layers of the same width and length in the proposed method where metal 1 connects these two layers. If we compare the NOR with the NAND gate, only the via connections (deep blue color layer) are different. According to the proposed approach, Fig 6 depicts the single gate that can use as NAND and NOR gates. With the camouflaging technique, modifications are done in layout, and with the application of the dummy contact concept, the whole design gets a unified shape.

## XOR design with camouflaged universal gates

Fig 7 shows the camouflaged XOR gate utilizing the universal gates. This gate is selected to see the effects of the circuit designed with the proposed camouflaged concept.

## Library development for IC camouflaging

In this work, metal routing-based camouflaging is applied. An attempt to design similar structured universal gates is made. As other logic gates are possible to design with only two universal gates NAND and NOR, we implement these two gates with camouflaging technique. An XOR gate is designed with the proposed technique to compare with the conventional design. Therefore, NMOS, PMOS, NAND, NOR, and XOR are the five designs available in the proposed library. As the EDA tool's training version is used, there could be deviations from the practical accuracy. On the other hand, a professional library is expensive. An entire library, including all the other logic gates, may be developed gradually. Each design has a schematic design to check functionality and its corresponding layout design to check physical functionality.

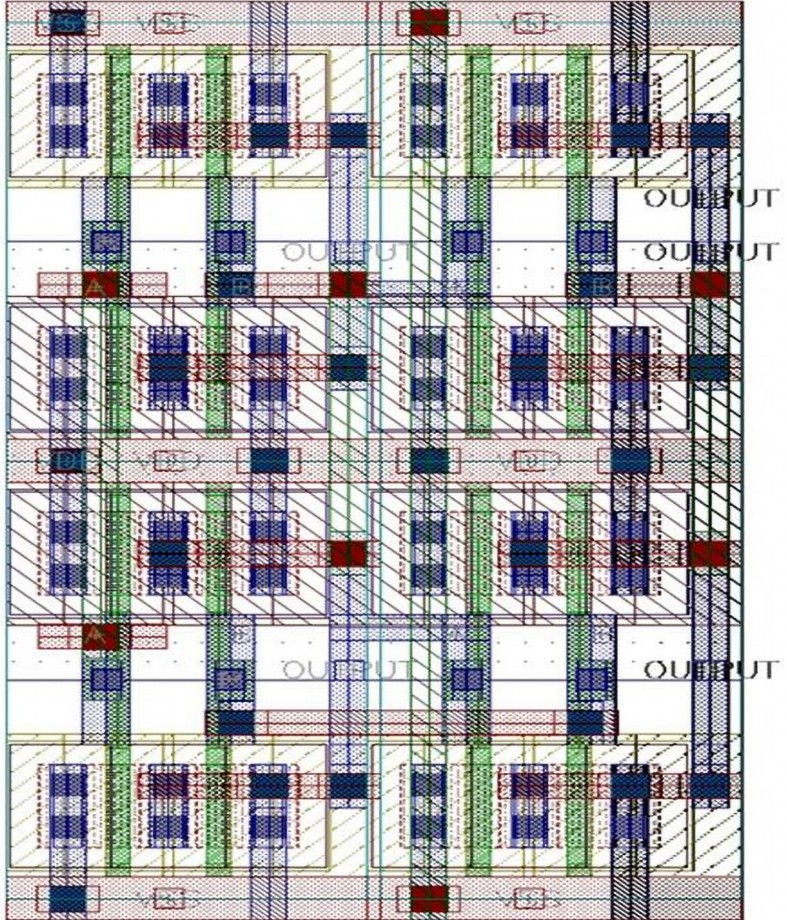

**Fig 7. XOR gate using camouflaged universal gates.**

PMOS and NMOS have the typical structure, and the size is kept similar to avoid any complexity. Usually, the size of the PMOS is kept 1.2–2 times of the NMOS in the higher node technologies, specifically in 180nm. But, here, the size has been kept similar as it does not affect the final result. Sizing could be optimized for a better result in the professional library setup called Process Design Kit (PDK). NAND and NOR are designed with these PMOS and NMOS devices. Using these NAND and NOR gates, an XOR gate is implemented to check these proposed gates' effects compared to the conventional gates.

## Results

The proposed method designs five cells in the library: PMOS, NMOS, NAND, NOR, and XOR. PMOS and NMOS have a general structure, and the size is kept similar to avoid any complexity. The used tools for this work are enlisted in Table 1. A detail explanation of each tools is provided in the following subsections.

### Cadence virtuoso schematic editor

We use the schematic editor for designing gates and symbolic creation. This tool provides options and faster ways to import any component and design any complex circuit with proper

**Table 1. Tools used for implementation of camouflaged gates.**

| Task | Tool Name | Vendor |
|---|---|---|
| Schematic Design | Virtuoso Schematic Editor | Cadence |
| Symbol Creation | Virtuoso Schematic Editor | |
| Simulation | Virtuoso ADE | |
| Layout Design | Virtuoso Layout Editor | |
| Physical Verification | PVS / Assura | |
| RC Extraction | | |
| Post Layout Simulation | | |

and accurate annotations. A considerable number of libraries are available in it to start any design at an instant. Cadence Virtuoso allows some tabs to open at the same time. In addition, it will enable hierarchical design opportunities, and there is no limitation of the level of the hierarchy of a design.

## Virtuoso ADE

The ADE is a design environment by Cadence that enables all the settings to simulate any design. Any simulation with any setup can run and check data generated from a specific design.

## Cadence virtuoso layout editor

This tool allows fabrication quality physical design of the design. It usually generates physical components from schematic, and the designer can design any complex gate or chip using this tool. GDSII file generating from the layout view design is forwarded to the Fab to manufacture the wafer design. So, the practical application of any design that has been designed using this software is practical. Both flat and hierarchical design options are available in this software.

## Cadence assura or PVS

After completing the schematic and its corresponding layout design of any logic gates or complex circuit, it is essential to check whether both designs are functioning well, maintaining the quality, etc. Assura (Cadence older version) or PVS (Cadence newer version) tool is used in the industry to verify those designs. They compare the functionality of any design, find out an error regarding design rules, and calculate total power consumption, total delay, and overall performance.

## Designing

In the first step, the schematic design of the gates is done. Node connections and pin placements are appropriately made. A proper simulation is run to check the functionality and design errors. No error is found, and the result comes as expected. This simulation run is also called a Pre-Layout simulation. When the layout is involved in the whole design, simulation is called Pre-layout simulation. In Fig 8, the XOR schematic and its simulation outcomes are shown.

## Physical layout design and verification

In this step, the necessary setup is done to start the layout design, like pins naming and placement. The layout design of each cell is completed. In this step, proper Floor-planning, device

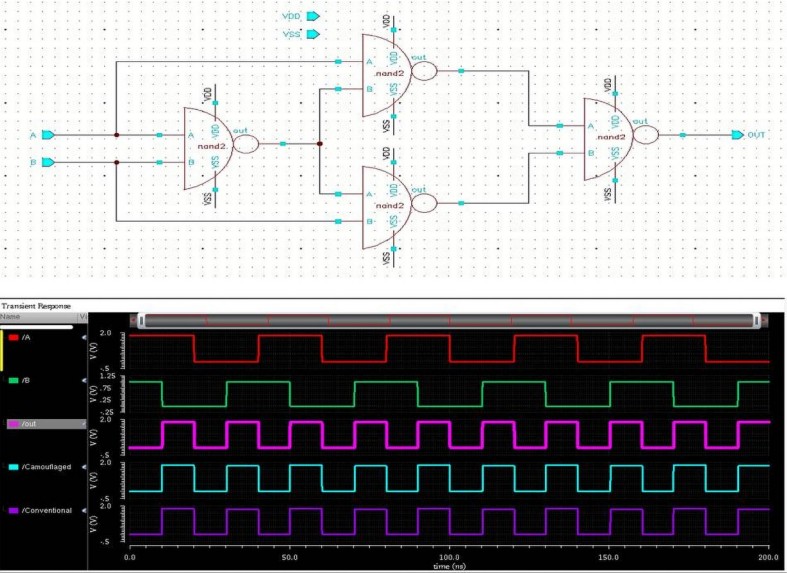

**Fig 8. XOR schematic circuit.**

placement are assured to confirm the maximum possible optimization. It can be seen from the schematic in Fig 9 that four NAND gates form the XOR logic gate. A single NAND is marked with a red-colored box. Four NAND gates are placed one by one to design an XOR. After the completion of the layout design, physical verification is done by checking the layout vs. schematic (LVS) and design rule checking (DRC). LVS is a comparison between the schematic and layout. The schematic of any gate should be matched with the layout of that gate. It ensures that the design is okay to move further, like post-layout simulation. In Fig 10, we see the

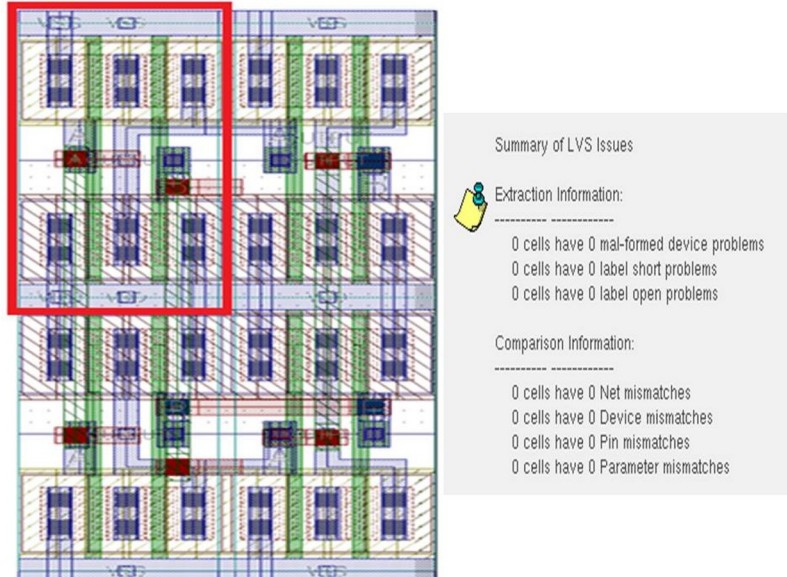

**Fig 9. XOR layout design and physical verification.**

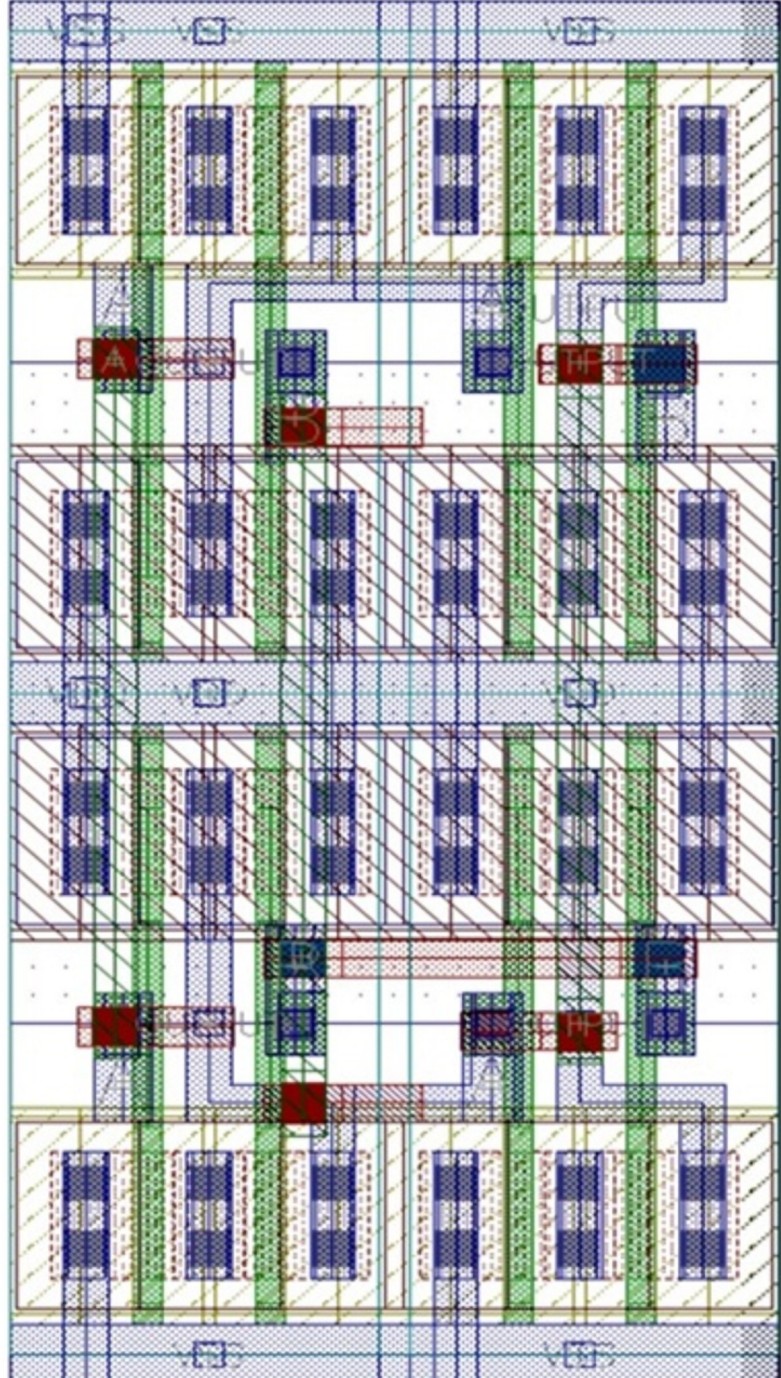

**Fig 10. XOR gate using camouflaged gates.**

numbers are zero. Zero means no error or warning found from the design resulting in LVS goes right.

In the final step, RC calculation is done to check delay and other performance. When the layout is designed, each and every layer has individual parasitics values that affect the

**Table 2. A comparison analysis between XOR conventional design and XOR camouflaged design.**

| Designs | Schematic | Layout | | Relative Difference |
|---|---|---|---|---|
| | | Conventional | Camouflaged | |
| Metal Hierarchy Used | N/A | M1, M2, M3 | M1, M2, M3 | nill |
| Area | N/A | 2.62 X 5.54 | 3.075 X 5.54 | 17.37% |
| Pre-Layout Simulation delay | 26.4743 ps | N/A | N/A | |
| Power (Static) | N/A | 21.2819 $\mu$W | 22.9714 $\mu$W | 7.94% |
| Power (Dynamic) | N/A | 43.5371 $\mu$W | 45.2047 $\mu$W | 3.83% |
| RC Extraction delay | N/A | 32.7456ps | 34.10262ps | 4.14% |

performance considering practical things. So, the delay is added due to these parasitic values. Fig 10 displays the complete XOR design using camouflaged universal gates. The post-layout simulation delivers its correctness. Table 2 shows the performance of the XOR gate.

The proposed method utilized the metal hierarchy of M1, M2, and M3 same as conventional practice. The delay, power, and area are measured from the list of tools provided in Table 1. A relative difference of the three parameters is shown in Table 2. The relative area difference is 17.37% ([3.075-2.62]/2.62), where power and delay are 3.83% and 4.14%, respectively.

## Comparison

A comparison of the proposed work with other related works is presented in Table 3. It compares area overhead, performance, and dynamic power values under various technologies. The proposed work outperforms the reported results in less area overhead. In terms of power consumption, the proposed work is outperformed except of [30]. The overall power consumption (-1.916*Power) and delay (-0049*Delay) of camouflaged gates in [30] is less compared to conventional gates as indicated by the minus sign. This is due to using an 80-k pull-up resistance in 2V condition. This additional resistance delivers a smaller current due to open and short circuit components in the network. Besides, the smaller value of delay results from shorter transitions due to a little voltage swing between logic-1 and logic-0. In [29], the power (0.2496 vs 0.0544) and delay (0.19 vs 0.041) is larger than that of the proposed approach though they used 45nm technology.

The area overhead in [30] is 25.5% (0.255*Area), which is higher than that of the proposed work, which is 17.37% (0.174*Area). Therefore, the area overhead should be considered in this context while deciding how many standard gates in the circuit are to be replaced with the camouflaged cell. That means in our proposed technique, the same level of security delivers with less area overhead.

**Table 3. A comparison among various camouflaged cells (NAND-NOR-XOR) under different technology scaling.**

| Work | Technology | power difference | Performance | Area overhead |
|---|---|---|---|---|
| [16] J. Rajendran et al | 180 nm | 5.5Xpower | 1.6Xdelay | 4Xarea |
| [28] N. Akkaya et al. | 65 nm | 9.2 X power | 6.6Xdelay | 7.3Xarea |
| [29] Patnaik S. et al. | 45 nm | 0.2496 X Power | 0.19 X Delay | 0.325 X Area |
| [30] Erozan AT. et al. | 200 um | -1.916 X Power | -0.049 X Delay | 0.255 X Area |
| This work | 180 nm | 0.0544 X Power | 0.041 X Delay | 0.174 X Area |

## Conclusion

An efficient and cost-effective IC camouflaging technique is essential for securing IC from reverse engineering. This work's main contribution is that a layout-level method for a library of universal gates is proposed to hide the circuit functionality where synthesizing circuits with logic cells. Still, with various functionalities, the functionality of original circuits cannot be extracted using physical RE. The developed gates are simple but highly efficient, low power, and low area overhead. A comparison with existing research shows that it outperforms all others. In the future, the library will be enriched with more primitive gates and will be open-source for the ASIC designer.

## Acknowledgments

The authors gratefully acknowledge to Institute of Information and Communication Technology (IICT) of Bangladesh University of Engineering and Technology (BUET) to conduct this research using all kinds of facilities of the embedded system Laboratory in the institute.

## Author Contributions

**Conceptualization:** Md. Liakot Ali.

**Data curation:** Fakir Sharif Hossain.

**Formal analysis:** Md. Liakot Ali.

**Investigation:** Md. Ismail Hossain, Fakir Sharif Hossain.

**Methodology:** Md. Ismail Hossain, Fakir Sharif Hossain.

**Resources:** Fakir Sharif Hossain.

**Software:** Md. Ismail Hossain, Fakir Sharif Hossain.

**Supervision:** Md. Liakot Ali.

**Writing – original draft:** Md. Liakot Ali, Fakir Sharif Hossain.

**Writing – review & editing:** Md. Liakot Ali, Fakir Sharif Hossain.

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
