## [Decision Letter · Decision Letter 0]

27 May 2021

PONE-D-21-04892

Area Efficient Camouflaging Technique for Securing IC Reverse Engineering

PLOS ONE

Dear Dr. Ali,

Thank you for submitting your manuscript to PLOS ONE. After careful consideration, we feel that it has merit but does not fully meet PLOS ONE’s publication criteria as it currently stands. Therefore, we invite you to submit a revised version of the manuscript that addresses the points raised during the review process.

Specially, one motivation paragraph is required in the revised version. Furthermore, innovations of the paper are not well presented.

We look forward to receiving your revised manuscript.

Kind regards,

Hua Wang

Academic Editor

PLOS ONE

Journal Requirements:

3. Thank you for stating the following financial disclosure:No

4. Thank you for stating the following in your Competing Interests section:No Competing Interests

5. Please amend the manuscript submission data (via Edit Submission) to include authors Fakir Sharif Hossain and  Md. Ismail Hossain.

Additional Editor Comments:

One paragraph of motivation for the paper is required.

Reviewers' comments:

Reviewer's Responses to Questions

**Comments to the Author**

1. Is the manuscript technically sound, and do the data support the conclusions?

Reviewer #1: Yes

Reviewer #2: Yes

2. Has the statistical analysis been performed appropriately and rigorously? 

Reviewer #1: Yes

Reviewer #2: Yes

3. Have the authors made all data underlying the findings in their manuscript fully available?

Reviewer #1: Yes

Reviewer #2: Yes

4. Is the manuscript presented in an intelligible fashion and written in standard English?

Reviewer #1: Yes

Reviewer #2: Yes

5. Review Comments to the Author

Reviewer #1: This paper explores IC obfuscation method by using true and dummy contact method. The authors have developed a opensource obfuscated standard cell library that can be used to replace some of the standard cells representing critical logic to protect the IP from reverse engineering. Overall the idea was interesting but heavily depends on the fab.

Reviewer #2: The study proposed a cost-effective camouflaging technique for two universal get NAND and NOR. It also proposed an open-sourced library with metal routing and dummy contact techniques. The methodology, explanation, and experimental evaluation are well written. The theoretical explanation of how it takes minimal overhead is also satisfactory.

However, as the authors claim the proposed method outperforms the existing approaches with the experimental result, they should detail explain how fairly comparison takes place with different technology in Table-3 (even it compares in ratio or percentage of times) in terms of power, performance, and area?

Also, in the comparison section, the explanation needs to explain more clearly from the reader's point of view. Like

1) how the values in line 329, 335, and 336 are drives from Table- 3?

2) more detail of the experimental setup for measuring the value of Table2,3 ( detail of Table-1)

3)A traditional small-scale benchmark can also be considered for Table-3.

6. PLOS authors have the option to publish the peer review history of their article (what does this mean?). If published, this will include your full peer review and any attached files.

Reviewer #1: No

Reviewer #2: No

---

## [Author Response · Author response to Decision Letter 0]

9 Jul 2021

Response to Reviewers

Editor Comments: 

Address: We have rechecked thoroughly the manuscript and corrected as per Plos template that is provided.

2. Please include captions for your Supporting Information files at the end of your manuscript, and update any in-text citations to match accordingly. Please see our Supporting Information guidelines for more information: http://journals.plos.org/plosone/s/supporting-information

Address: There is no supporting information. All data are within the manuscript. 

3. If you did not receive any funding for this study, please state: “The authors received no specific funding for this work.”

Address: We have added in the cover letter.

4.Please complete your Competing Interests on the online submission form to state any Competing Interests. If you have no competing interests, please state "The authors have declared that no competing interests exist.", as detailed online in our guide for authors at http://journals.plos.org/plosone/s/submit-now.

Address: We have added in the cover letter (The authors have declared that no competing interests exist).

5. Please amend the manuscript submission data (via Edit Submission) to include authors Fakir Sharif Hossain and Md. Ismail Hossain.

Address: Thank you for the comment. We have added two more authors in the list.

Address: Thank you for the comment. We have checked thoroughly and found ref. [12] should be replaced by ref. [40] in line 84. 

Additional Editor Comments:

Address: Thank you so much for your valuable comments. We have added one more section named as “Related work and Motivation” in Line 42-116. 

Reviewer comments

1) In line 30-32 the author mentioned “Another way of making camouflaged cells is to control the doping concentration in the drain area of a MOS gate so that the NMOS remains ON and the PMOS always remains OFF state, resulting in a NAND gate functioning as an inverter [12]”

>> Here the author mentioned an obfuscation process of disguising an inverter as a NAND gate using doping concentration. But ref [12] does not provide any such methods. The referencing seems wrong and must be updated.

Address: Thank you so much for the comments. We mistakenly did the wrong referencing. Sorry for the mistake. We have revised the reference and put the appropriate ref. of [40] in line 83-84. “Another way of making camouflaged cells is termed as `covert gate' that leverages doping and dummy contacts to create camouflaged cells [40].”

2) In line 90-92 From the designers or the defenders’ point of view, some decisions are made, such as the number of camouflaged cells; the design affords what structure the camouflaged cells is to use in the design in what part of the design it may use these camouflaged cells.

>> This sentence contains grammatical error and need to be rewritten.

Address: Thank you so much for the comments. We have corrected the sentence in line 148-150 such as “From the defenders' point of view, some decisions on camouflaged cells are made. The number of camouflaged cells the design affords with the structural usage and in what part of the design it may use these camouflaged cells.”

Reviewer #1: This paper explores IC obfuscation method by using true and dummy contact method. The authors have developed a open source obfuscated standard cell library that can be used to replace some of the standard cells representing critical logic to protect the IP from reverse engineering. Overall the idea was interesting but heavily depends on the fab.

Address: Thank you for the comment. We wholly agreed with you sir. In our future scope we may go for fab and also may try to improve the proposed library.

Reviewer #2: The study proposed a cost-effective camouflaging technique for two universal get NAND and NOR. It also proposed an open-sourced library with metal routing and dummy contact techniques. The methodology, explanation, and experimental evaluation are well written. The theoretical explanation of how it takes minimal overhead is also satisfactory.

However, as the authors claim the proposed method outperforms the existing approaches with the experimental result, they should detail explain how fairly comparison takes place with different technology in Table-3 (even it compares in ratio or percentage of times) in terms of power, performance, and area?

Also, in the comparison section, the explanation needs to explain more clearly from the reader's point of view. Like

1) how the values in line 329, 335, and 336 are drives from Table- 3?

2) more detail of the experimental setup for measuring the value of Table2,3 ( detail of Table-1)

3)A traditional small-scale benchmark can also be considered for Table-3.

Address: Thank you for your comments. We have tried to address in our best.

1) For the more clarifications, we have rewritten the Comparison section in line number 379. We put some detail in this section according to TABLE-3 (line-381-391). Hopefully, the measuring value from Table-3 can be understandable.

2) We have added a detail explanation of tools used, experimental setup and how to calculate them in the result section from line 323 to 347.

3) We highly appreciate you direction on “A traditional small-scale benchmark can also be considered for Table-3”. We apologize, currently; we are unable to add them because it is out of scope right now. We do not have such data; however, we might consider it in our futu

---

## [Decision Letter · Decision Letter 1]

8 Sep 2021

Area Efficient Camouflaging Technique for Securing IC Reverse Engineering

PONE-D-21-04892R1

Dear Dr. Ali,

We’re pleased to inform you that your manuscript has been judged scientifically suitable for publication and will be formally accepted for publication once it meets all outstanding technical requirements.

Kind regards,

Hua Wang

Academic Editor

PLOS ONE

Additional Editor Comments (optional):

Reviewers' comments:

Reviewer's Responses to Questions

**Comments to the Author**

1. If the authors have adequately addressed your comments raised in a previous round of review and you feel that this manuscript is now acceptable for publication, you may indicate that here to bypass the “Comments to the Author” section, enter your conflict of interest statement in the “Confidential to Editor” section, and submit your "Accept" recommendation.

Reviewer #1: All comments have been addressed

Reviewer #2: All comments have been addressed

2. Is the manuscript technically sound, and do the data support the conclusions?

Reviewer #1: Yes

Reviewer #2: Yes

3. Has the statistical analysis been performed appropriately and rigorously? 

Reviewer #1: Yes

Reviewer #2: Yes

4. Have the authors made all data underlying the findings in their manuscript fully available?

Reviewer #1: Yes

Reviewer #2: Yes

5. Is the manuscript presented in an intelligible fashion and written in standard English?

Reviewer #1: Yes

Reviewer #2: Yes

6. Review Comments to the Author

Reviewer #1: Authors have addressed all my queries and updated the manuscript accordingly. I whole heartedly thank authors for their hard work and hope they will continue work on the topic to cover all the limitations.

Reviewer #2: (No Response)

7. PLOS authors have the option to publish the peer review history of their article (what does this mean?). If published, this will include your full peer review and any attached files.

Reviewer #1: **Yes: **Shafaitul Islam Surush

Reviewer #2: No

---

## [Editor Report · Acceptance letter]

27 Oct 2021

PONE-D-21-04892R1 

Area Efficient Camouflaging Technique for Securing IC Reverse Engineering 

Dear Dr. Ali:

I'm pleased to inform you that your manuscript has been deemed suitable for publication in PLOS ONE. Congratulations! Your manuscript is now with our production department. 

Kind regards, 

on behalf of

Dr. Hua Wang 

Academic Editor

PLOS ONE